# OpenReview forum: "Comparison requires valid measurement: Rethinking attack success rate comparisons in AI red teaming"
_NeurIPS.cc/2025/Position_Paper_Track — NeurIPS 2025 Position Paper Track_

### Official Review · Reviewer_fgin · 2025-08-03

**Significance:** 4
**Presentation:** 4
**Rating:** 9
**Confidence:** 4

**Summary:**

This position paper maintains that the conclusions involving AI system safety deriving from the comparisons of Attack Success Rate (ASR) are often fundamentally flawed. It identifies a conceptual incoherence where the comparability of the ASRs used is di erent in the underlying quantities (estimands) being evaluated.

This paper concerns itself mostly in providing a formal measurement framework of the ASRs that is di erentiated from the social sciences, which aids in conceptualizing safety. This framework also separates high-level safety estimands (theoretical success probability) from the ASR used.

ASR comparisons to be more meaningful, the paper emphasizes the need for researchers to explicitly describe their estimands. It states that ASR measurement relies on providing precise criteria for ASR which in turn leads to poorly-defined measurement.

**Strengths:**

This paper is an exemplarity of clarity, logic, and persuasion in its argumentation. It constructs a new and useful measurement framework taken from social science to identify deep-seated issues in the AI red teaming evaluation paradigm. The argument is cohesive and meticulously crafted, blending robust conceptual reasoning, insightful case studies of pivotal published work, and targeted empirical evidence that vividly illustrates the core claims. Why this topic is critical and timely for the NeurIPS community is that it concerns the validity of the measurements that were used to justify the AI safety claims. The paper is not a mere critique; it is proactive by offering constructs aimed at helping researchers by providing them the theories and actionable guidance needed to perform more meaningful evaluations.

**Weaknesses:**

The debate around 'AI red teaming' tends to use the term too broadly. This term should be scoped to the focus on automated and quantitative jailbreaking assessments. While the paper’s theoretical framework is coherent, it must confront the practical challenge of defining ‘the oracle’ for complex, contested biases or manipulations, which is essential to its proposed process. The primary alternative position addressed is the status quo of ad-hoc evaluation. The paper could further construct its argument to consider other evaluation philosophies, those which aim to derive a diverse portfolio of exploitable weaknesses as opposed to a single, easily comparable ASR or those which privilege qualitative ‘existence proofs’ as the primary goal of red teaming.

**Questions:**

Your framework provides strong justification for comparing conceptually coherent estimands. In cases of more qualitative, exploratory forms of red teaming where a single ASR does not serve as a serting a diverse portfolio of "unknown unknown" vulnerabilities, how would you apply it? Is it possible that a system’s safety can be better described as a vector of coherent ASRs for different threat models than a single score?

The oracle success criterion s remains a powerful theoretical construct. In the case of subtle biases or more complex, contested harms like manipulation, what practical steps or best practices can be taken to develop a defensible systematization of s in the absence of a true oracle, and do so, even if imperfectly?

**Alternative Position:**

Yes, and alternative positions are well-considered and addressed by the argument

**Author Identification:**

No.

**Context:**

4

**Discussion:**

4

**Ethics:**

["NO or VERY MINOR ethics concerns only"]

**Position:**

Yes, the paper argues for or against a position related to machine learning.

**Support:**

4

**Thoroughness:**

4

---

### Official Review · Reviewer_Zjxv · 2025-08-15

**Significance:** 3
**Presentation:** 3
**Rating:** 6
**Confidence:** 4

**Summary:**

The paper critiques the common practice of comparing "Attack Success Rates" (ASRs) in AI red teaming, arguing that such comparisons are often flawed and don't provide meaningful insights into a system's safety. The authors propose that for ASR comparisons to be valid, they must satisfy two conditions: conceptual coherence, meaning the attacks and outcomes being measured are truly comparable, and measurement validity, ensuring the ASR accurately reflects the intended concept. The paper details how current red teaming efforts often violate these conditions through poor quality prompts, unreliable judges, and misleading "apples-to-oranges" aggregations, leading to misleading conclusions about AI safety. It uses recent "jailbreak" studies to illustrate these failures and recommends stricter definitions, more valid threat models, and better evaluation methods to improve the rigor of AI safety testing.

**Strengths:**

The paper presents a strict, formal framework for determining when Attack Success Rates can be reliably compared. It offers both the theoretical foundation and real-world examples to show the common mistakes in current research. The examples are specific, well-explained, and directly relevant to the topic. The paper's recommendations, if adopted, have the potential to significantly improve the accuracy of metrics used in AI red teaming.

**Weaknesses:**

The paper is overly long and repetitive, relying heavily on examples of failures without acknowledging when Attack Success Rate comparisons might be useful. It also fails to address practical counterarguments, making its theoretical framework seem too complex for the issues it's discussing.

**Questions:**

How do the authors propose we retroactively correct for issues like conceptual incoherence or judge bias in those already-published results?

**Alternative Position:**

No

**Author Identification:**

No.

**Context:**

4

**Discussion:**

4

**Ethics:**

["NO or VERY MINOR ethics concerns only"]

**Position:**

Yes, the paper argues for or against a position related to machine learning.

**Support:**

4

**Thoroughness:**

4

---

### Official Review · Reviewer_hX8c · 2025-09-01

**Significance:** 2
**Presentation:** 2
**Rating:** 5
**Confidence:** 3

**Summary:**

This paper argues that ASR (attack success rates) are compared for AI red teaming, researchers need to be more careful thinking about what they are measuring and that this is constant across comparisons, and that this measurement reflects what you care about. The paper proposes precise guidelines to avoid potential problems when comparing ASRs.

**Strengths:**

- takes a concrete example where a paper does not make valid comparisons

- underlying position is surely true, you should keep estimand constant when making comparisons!

- gives clear guidelines for researchers

**Weaknesses:**

- estimand bit is overly formal for a fairly basic point should be majorly streamlined to make the intuitive point
- Only a single example presented on when researchers are not holding the estimand constant, hard for me to tell how common this problem is.

**Questions:**

1. Is there any evidence this is a widespread problem beyond the single example you go over? If you could do some sort of systematic review of ASR's being problematically compared that would update my score.
2. Doesn't this problem hold for all ML comparisons (e.g. benchmarking?)

**Alternative Position:**

Yes, and alternative positions are well-considered and addressed by the argument

**Author Identification:**

No.

**Context:**

4

**Discussion:**

3

**Ethics:**

["NO or VERY MINOR ethics concerns only"]

**Position:**

Yes, the paper argues for or against a position related to machine learning.

**Support:**

3

**Thoroughness:**

3

---

### Official Review · Reviewer_y9wV · 2025-09-06

**Significance:** 3
**Presentation:** 3
**Rating:** 5
**Confidence:** 3

**Summary:**

This article points out that many current red team evaluation comparisons based on attack success rate (ASR) often draw untenable conclusions due to inconsistent comparison objects (apples to oranges) or insufficient measurement validity. Drawing on social science measurement theory and inferential statistics, the author proposes two conditions that guarantee comparability:

1. Conceptual coherence—the "estimands" being compared must be consistent. 2. Measurement validity—ASR, as a measurement, should truly reflect the claimed "system security/attack effectiveness." Focusing on jailbreak evaluations, the article systematically deconstructs how different aggregation methods (e.g., one-shot vs. Top-1/Best-of-K/Any-from-T) alter the "quantity being measured," rendering direct comparisons impossible across studies or methods. Using the example of GE vs. GCG, the article illustrates that comparing "Top-1" (392 times) with "one-shot" (1 time) is inherently incomparable. The authors also demonstrate that errors in the judgement (LLM-as-judge) can cause model rankings to reverse, demonstrating that simply using a unified judgement is not sufficient to ensure valid comparisons.

**Strengths:**

The problem definition is clear and universal: It elevates "ASR comparison" to the level of "estimand comparison" and, using a medical trial analogy, clearly distinguishes between three types of conclusions: descriptive, inferential, and evaluative. The logic is self-consistent and easily transferable to other evaluation scenarios. The methodological foundation is solid: It incorporates the social science measurement theory framework (concepts - systematized concepts - measurement tools - observables) and the "probabilistic threat model," elevating red team evaluation from an empirical practice to a reasoned measurement process. The impact of "aggregation method → ​​measured object" is revealed in depth: the "ontological changes" of indicators such as one-shot, Best-of-K, Top-1/Any-from-T are systematically sorted out, and the inevitability of "increase in option set size → monotonically increase in ASR" is clarified, avoiding the mistaken interpretation of sampling strategy differences as method advantages and disadvantages. The empirical criticism is convincing: using the comparative example of GE vs. GCG, it accurately points out the incomparability of "Top-1 (392) vs. one-shot (1)";

**Weaknesses:**

Narrow scope: Almost all analysis focuses on jailbreaking/ASR and prompt-based interactions; attacks requiring weighted access or fine-tuning of interfaces are simply acknowledged as "not covered." This limits the generalizability of the conclusions to other red teaming strategies.

Narrow empirical scope and single subject: Reproduction experiments primarily focus on LLAMA 2 (7B/13B), a fixed set of 100 MaliciousInstruct prompts, and 49 decoding configurations, resulting in insufficient sample and model diversity.

The experimental setup may deviate from common deployments: To demonstrate that multiple Top-1 sampling can improve ASR, the authors employed high-entropy decoding and high temperatures (even up to 1.5 and 2.0). While this demonstrates statistical effects, its generalizability to conventional conservative decoding is questionable.

**Questions:**

Same as the weakness.

**Alternative Position:**

Yes, and alternative positions are trivial straw-man arguments

**Author Identification:**

No.

**Context:**

3

**Discussion:**

3

**Ethics:**

["NO or VERY MINOR ethics concerns only"]

**Position:**

Yes, the paper argues for or against a position related to machine learning.

**Support:**

2

**Thoroughness:**

4

---

### Note · Authors · 2025-09-05

**1-10 Additional Comments:**

We would love to see a position paper track at NeurIPS succeed!  We are optimistic that with the benefit of a longer planning horizon next year's track chairs would be able to improve the review process and author experience.  If appropriate improvements are made, we would be very happy to submit to the track again in future years.  If the review process and expected review quality remains unchanged from this year, then we would be unlikely to submit again.

**1-11 Submit Again:**

Unsure

**1-1 Submission Process:**

2

**1-2 Next Year:**

While the initial submission process was very smooth, we have been frustrated with the review process.  We understand that this was a difficult track to run and that everything came together last minute.  As authors with experience program chairing conferences on a fairly last-minute timeline, we really do sympathize with the challenges the track chairs must have encountered this year.

*Recommendations for next year*
- Set clear expectations for the review process as early as possible, ideally as part of the CFP, and then hold to that process.  It wasn't at all clear prior to late July that the position track papers would be reviewed on a different schedule, and the shifting dates were very difficult to manage as authors.  Please also be clear in advance about what kind of rebuttal/author response options will be made available.

- The review process should not treat position papers as work that doesn't require domain knowledge or technical expertise to review.  We saw the emails that were being sent out recruiting emergency reviewers, which could be interpreted as asserting that papers that require some level of domain knowledge to assess are "inscrutable."  Yet many of the best position papers are grounded in a deep engagement with technical issues that are going overlooked or broadly misunderstood by the research community.  Expertise is necessary in many cases to give papers the rigorous review they deserve.  We worry that downplaying the role of relevant expertise in reviewing position papers has the effect of framing position papers as “second class citizens” that do not require thorough and rigorous review.

**1-4 Interest:**

["Panel discussions with other position paper authors", "Structured debates on controversial topics"]

**1-5 Thoughtful:**

3

**1-6 Supportive:**

7

**1-7 Technical Aspects Versus Position:**

9

**1-8 Gate Keeping:**

10

**1-9 Camera Ready Changes:**

We intend to make the following revisions to the main text and appendix.  [xxxx] indicates the reviewer id whose comments most directly motivate the proposed change.

*Main text*

[hX8c] When providing examples of failures of conceptual coherence, we will explain that such failures are often difficult to identify in published research because researchers/practitioners often do not provide enough information to know precisely how ASRs are being calculated.  Even accompanying code and data often doesn’t contain the parts of the code where ASRs are calculated.  One of our recommendations (present in the existing conclusion) is that researchers/practitioners should define their estimands explicitly.

[hX8c] We will briefly note that our arguments apply beyond attack success rates and AI red teaming to other forms of evaluation as well. A natural place to do this is when we discuss other very recent literature on measurement theory and genAI evaluation.

[Zjxv] We will discuss why “retroactively correcting” for issues in already published studies is often not possible without rerunning the experiments.  In most cases the issue is that the data logged by authors is insufficient to support the analysis such a “retroactive correction” might require.  Indeed this is why in our paper we wound up needing to completely re-run experiments from previously published studies ourselves.

[fgin] We will provide additional discussion and examples of “the oracle” for complex contested conceptualizations of safety.

*Appendix*

[fgin] We will expand our related work section on AI red teaming to provide further discussion of how the term “AI red teaming” has come to be used very (arguably, overly) broadly.

[fgin] Following the measurement theory section, we will also provide additional discussion of qualitative uses of AI red teaming and what parts of the framework are relevant to qualitative evaluation methodologies (elements like how safety is conceptualized).

**3-1 Review Response1:**

hX8c

**3-2 Reaction To Review1:**

*Thoughtfulness*  We are deeply concerned that the review appears to ignore more than half of our contribution.  Our paper identifies two important criteria for meaningful comparison: *conceptual coherence* and *measurement validity*, and much of the paper and examples therein are devoted to measurement validity concerns.  Yet the review comments only on conceptual coherence.  The reviewer misstates our position as being only that researchers/practitioners “should keep the estimand constant when making comparisons,” missing the measurement validity piece altogether.

We understand the reviewer’s main concern to be that we provide "only a single example" of conceptual incoherence (incomparable estimands) in Section 4 of the paper.  Yet that section contains 3 examples (called “Parts”), drawn from two distinct papers.  We also note that identifying examples of conceptual incoherence requires knowing how, precisely, the authors computed the attack success rates they report.  This is seldom sufficiently well detailed in text, and is often left out of any accompanying code.  Indeed, one of our key recommendations is that researchers/practitioners “ensure conceptual coherence by *precisely defining estimands that can be meaningfully compared*”. This would make apples-to-oranges comparisons apparent to researchers/practitioners, and thus avoidable.

*Supportiveness* The reviewer appears to support the conceptual coherence part of our argument. We appreciate that the reviewer notes our position to be applicable to other comparisons in ML, such as benchmarking more broadly.

*Technical aspects vs position*. The reviewer focuses primarily on the position, but states that the technical level is overly formal for at least the conceptual coherence discussion in the paper.  We believe the framework we present is critical to fully expounding not only the conceptual coherence points, but also the measurement validity arguments.

*Gatekeeping*.  None noted.

**3-3 Review Response2:**

Zjxv

**3-4 Reaction To Review2:**

*Thoughtfulness*:  We are disappointed that in the weaknesses section the reviewer did not articulate their concerns in a manner we can meaningfully engage with in revising the work.  For instance, the reviewer states that the paper is “overly long and repetitive” and "“[the paper] also fails to address practical counterarguments" without pointing to any specific problematic repetition or giving any examples of a "practical counterargument".

We are also perplexed by the comment that we do not acknowledge “when ASR comparisons might be useful.“ In the abstract and introduction, explicitly pose the question: “When can attack success rates be meaningfully compared as reflections of relative system safety or attack method efficacy?”  As the reviewer states in their summary of our paper, we argue that for ASR comparisons to be meaningful we need conceptual coherence and measurement validity.  Comparisons that lack conceptual coherence or where the ASRs computed through experiments lack measurement validity are not meaningful and not useful.  ASRs that are that are conceptually coherent and are obtained through valid measurement are meaningful and useful.   Our framework and recommendations offer a clear path toward ensuring conceptual coherence and assessing measurement validity to ensure ASR comparisons are meaningful and useful.

*Supportiveness* We appreciate that the reviewer believes our paper does a good job of providing examples from published studies.  We are glad they believe our recommendations, if adopted, have the potential to significantly improve attack success rate comparisons obtained through AI red teaming.

*Technical aspects vs position*. The reviewer states that the theoretical framework we present is overly complex for the issues we are discussing.  This suggests an issue with the technical depth of the work (that there is perhaps too much) rather than the position taken.

*Gatekeeping*  None noted.

**3-5 Review Response3:**

fgin

**3-6 Reaction To Review3:**

*Thoughtfulness* The reviewer provides a very thoughtful review in which they offer both positive feedback and areas for improvement/questions that were clearly articulated and easy for us to engage with in revising the paper.  Several of the issues raised (e.g., how “AI red teaming” is a very broad term, and our very limited discussion of qualitative red teaming approaches) are ones that we had content on in earlier drafts of the manuscript but were ultimately cut for space.  The review helps us recognize that the paper’s audience may be quite interested in our discussion of those topics, so we will be sure to include it at least an appendix in a camera-ready.

*Supportiveness* The reviewer offers very strong support, noting that the paper is “not a mere critique” but also offers actionable guidance to move the field forward.

*Technical aspects vs position*. The reviewer focuses on our position.

*Gatekeeping*  None noted.

---

### Meta-Review · Area_Chair_vi4c · 2025-09-01

**Rating:** 7
**Confidence:** 4

**Strengths:**

+ The paper has a clear position and is overall well structured.

+ The paper provides a rigorous formal analysis of the proposed position.

**Weaknesses:**

- The paper can benefit from empirical support of the proposed position.

**Questions:**

What would be the practical suggestions and best practices for effective red-teaming if ground truth is not available?

**Ethics:**

No ethical issue found.

**Thoroughness:**

4

---

### Decision · Program_Chairs · 2025-09-26

Accept